# Simple Wireless Impedance Pneumography System for Unobtrusive Sensing of Respiration [note 1]

**DOI:** 10.3390/s20185228

**Published:** 2020-09-14

**Authors:** Pablo Aqueveque, Britam Gómez, Emyrna Monsalve, Enrique Germany, Paulina Ortega-Bastidas, Sebastián Dubo, Esteban J. Pino

**Affiliations:** 1Electrical Engineering Department, Faculty of Engineering, Universidad de Concepción, Edmundo Larenas 219, Concepción 4070409, Chile; pablo.aqueveque@biomedica.udec.cl (P.A.); britam.gomez@biomedica.udec.cl (B.G.); emyrna.monsalve@biomedica.udec.cl (E.M.); enrique.germany@biomedica.udec.cl (E.G.); 2Kinesiology Department, Faculty of Medicine, Universidad de Concepción, Chacabuco 1401, Concepción 4070409, Chile; portegab@udec.cl (P.O.-B.); sedubo@udec.cl (S.D.)

**Keywords:** impedance pneumography, respiration, wireless

## Abstract

This extended paper presents the development and implementation at a prototype level of a wireless, low-cost system for the measurement of the electrical bioimpedance of the chest with two channels using the AD5933 in a bipolar electrode configuration to measure impedance pneumography. The measurement device works for impedance measurements ranging from 1 Ω to 1800 Ω. Fifteen volunteers were measured with the prototype. We found that the left hemithorax has higher impedance compared to the right hemithorax, and the acquired signal presents the phases of the respiratory cycle with variations between 1 Ω, in normal breathing, to 6 Ω in maximum inhalation events. The system can measure the respiratory cycle variations simultaneously in both hemithorax with a mean error of −0.18 ± 1.42 BPM (breaths per minute) in the right hemithorax and −0.52 ± 1.31 BPM for the left hemithorax, constituting a useful device for the breathing rate calculation and possible screening applications.

## 1. Introduction

Bioimpedance analysis has been applied in the measurement of body composition and as a measurement tool in healthcare [1]. Nowadays, there are more uses of bioimpedance techniques in biomedical applications thanks to its low cost, secure handling, and the improvement of electronic components [2,3]. This has motivated many researchers to develop portable and wearable electric bioimpedance measurement systems [4] for personal and home monitoring [5,6,7,8,9], but whose applications focus on aesthetics, like body composition, rather than physiological aspects [1,10,11,12,13].

Since the 1960s, bioimpedance techniques have not only been dedicated to the evaluation of body composition, but have also been incorporated as an assessment technique in respiratory care [14,15], obtaining valuable information on the state of tissues or internal organs [16,17,18,19,20]. In recent years, the clinical use of bioimpedance acquisition has gained significant interest in pneumology due to its applicability in the monitoring of variables related to pulmonary function like tidal volume distribution and respiratory rate [21,22,23]. This area includes measurement systems such as electrical impedance tomography (EIT) and Impedance pneumography (IP).

EIT is a noninvasive, radiation-free clinical imaging tool to monitor the distribution of ventilation in real-time and at the bedside of patients [21,22,23]. These advantages are particularly valuable in hospitalized patients, in whom their clinical status limits the possibility to transfers to support clinical services and could be a preliminary alternative to assess continuous, noninvasive, and radiation-free information of lung function. [3,24].

On the other hand, IP is a commonly used technique for monitoring respiratory rate [24]. This assessment is based on the fact that during breathing, the chest generates an impedance that is composed by two values: a relatively constant mean impedance (RB) of ~500 Ω, which varies depending on the position of the electrodes and the subject’s skin characteristics [25] (this measure at 50 kHz is a commonly used frequency for thoracic impedance measurements [26]), and a variable component (△R) that represents the relative changes in lung electrical impedance during the respiratory cycle, with a mean variation respect RB of ±2.7 Ω in normal subjects with normal breathing [27]. The IP approach does not consider image reconstruction of the lung region like EIT but, by sensing and analyzing spatio-temporal structural signatures in the impedance distribution, more information can be extracted with a simpler set-up and less computational processing requirements [28].

During inspiration, the decrease in pleural pressionas result of inspiratory muscle contraction generates a negative intralveolar pressure, bellow atmospheric pressure, that finally moves air to the lung, expanding the thorax and inducing a decrease in tissue conductivity. Furthermore, the conductivity increases due to thoracic expansion. These changes produced during inspiration imply an increase in the electrical impedance relative to the basal impedance, which can be related to the changes in volume [14,15,29,30,31]. These systems, which allow the continuous evaluation of the patient on the bedside, have proven to be an important tool in the monitoring of the respiratory system in pathologies such as distress and mechanically ventilated patients [22,23].

While the bioimpedance measurement technique is simple, hardware development can be challenging. The AD5933 (Analog Devices, Norwood, MA, USA) [32] is the first commercially available Impedance Analyzer (IA) that allows the implementation of small size devices for electrical bioimpedance measurements and is the most used one [4,33,34]. While there are lower-cost integrated systems to measure impedance based on the AFE4300 that are focused on body composition applications, and others like the ADS129x that have an integrated system for impedance respiration measurement, the AD5933 has the advantage of being a chip that allows you to make frequency sweep measurements at different voltages or excitation currents easily, offering wide versatility for different applications [4]. However, this does not limit the aforementioned circuits from being used for similar applications but with similar challenges presented in this development [8,35].

In this extended work from the conference paper in [36], we present the development and implementation, at a prototype level, of a portable wireless system to assess variations in thoracic electrical bioimpedance (TEB) during breathing, using the AD5933. This study also reviews the main characteristics and challenges encountered during the measurement of TEB in healthy volunteers with the proposed system. We detail the safety considerations used during the design of the device, the electrode configuration to measure the impedance, the calibration procedure of the device, improved schemes of the measurement set-up, and improved results obtained with regard to the conference paper. We also profusely discuss the importance of this development, especially in applications for home monitoring as a point of care (POC) device to screen lung anomalies.

## 2. Impedance Pneumography Measurement Device

### 2.1. Safety Considerations

The effects of the electrical current on the body depend on current intensity, contact time, body impedance, physiological conditions, and current frequency.

The maximum intensity that a person can safely withstand, regardless of the exposure time, is called the safety threshold. Different studies carried out on the human body have established this limit at 1 mA${}_{peak}$ at frequencies between 1 kHz and 100 kHz [37], which is under the perception threshold [38]. Furthermore, this current limit ensures that the designed equipment is safe for the patient even when malfunctioning [37].

Regarding the frequency of the alternating current, if the current frequency increases it could decrease the risks of ventricular fibrillation. However, the risk of burns increases [39].

The TEB is considered safe for several factors. The first is that at a frequency of 50 kHz [26], commonly used in monofrequency equipment, it does not stimulate electrically excitable tissue such as nerves or heart muscles. A second factor is the absence of reports of TEB-induced adverse events even when the number of people who undergo these measurements is high. Furthermore, a third factor is that small current, less than 1 mA, is used. In the case of wireless and portable devices, the use of batteries or low-voltage power sources significantly reduces the risk of accidents [40].

### 2.2. Electrode Configuration

Changes in chest electrical impedance caused by respiration can be measured using either a two-electrode (bipolar) or a four-electrode (tetrapolar) configuration [26,41], as shown in Figure 1.

In both configurations, an alternating current under the perception threshold is injected into the chest at a high frequency (50–100 kHz) using surface electrodes. The current causes a potential difference that is measured between the electrodes. The equivalent resistance is defined as the ratio between the voltage in the receiving electrode(s) and the current applied to the tissue.

Although the tetrapolar configuration provides better measurements than the bipolar configuration [26], in this development, the bipolar configuration is preferred as it requires fewer electrodes, which reduces the user’s discomfort and provides the necessary information to obtain the desired respiratory waveforms.

### 2.3. Hardware Design

The hardware used to acquire the TEB variations are divided into three stages, considering performance and safety considerations [38,39,42,43]: control, excitation signal generation, and distribution and acquisition. A general diagram of the proposed design is shown in Figure 2.

#### 2.3.1. Control Stage

An ATmega328 microcontroller (Microchip Technology Inc., Chandler, Arizona, USA) was used to control the impedance analyzer (Analog Devices, Norwood, MA, USA) functioning and the analog multiplexer commutation. The system stores the measured data and sends them through serial communication to an analysis and processing platform through a Bluetooth module.

#### 2.3.2. Excitation Signal Generation Stage

An excitation voltage of 1 V${}_{pp}$ was generated using the AD5933 at a fixed frequency of 50 kHz to ensure a safe current circulation through the tissues and avoid muscle and cardiac stimulation [39].

A voltage-controlled current source (VCCS) of 2 mA${}_{pp}$ was implemented using an operational amplifier TL081 (Texas Instruments, Dallas, Texas, USA) in a “load-in-the-loop” configuration [44] to control the current flow into the body, which was applied to the body using Ag/Cl electrodes of 40 mm × 35 mm (3M, St. Paul, MN, USA) in a bipolar configuration [41]. The operation of the current source remains constant in the range from 7 Ω to 2 Ωk, delivering 706.8 uA${}_{RMS}$ with no offset.

A load-in the-loop current source was considered because of its simple design and disadvantages compared to other sources that do not affect this development’s claims when using a single frequency of 50 Khz [45]. We highlight that the intention is not to obtain precise impedance measurements for the generation of images, but rather to obtain its variations and extract characteristics that could provide essential parameters in the future using a simplified system.

#### 2.3.3. Distribution and Acquisition of the Excitation Signal

A dual-channel analog multiplexer MC14052B (ON Semiconductor, Phoenix, AZ, USA) was used to deliver the excitation current to the chest because of the single excitation output from the impedance analyzer. An instrumentation amplifier INA128p (Texas Instruments, Dallas, TX, USA) measures the voltage between the electrodes and is transformed by the AD5933 to digital values proportional to the real and imaginary parts that represent the magnitude and phase of the measured impedance.

### 2.4. Implementation

#### 2.4.1. Bioimpedance Pcb

Figure 3 shows the implemented PCB for the TEB device. The system is composed of (1) a ATmega328 microcontroller, (2) a AD5933 impedance analyzer, (3) an analog front-end stage, (4) a dual-channel analog multiplexer, (5) a bioimpedance electrodes connector, (6) a ±3 V dual voltage regulator, and (7) Bluetooth 3.0.

The TEB device was powered from two 3.7 V/500 mAh LiPo batteries.

#### 2.4.2. Vccs Testing

The behavior of the current source was evaluated using a simple electrical skin model, with R1 = 1 Ωk, C1 = 0.47 uF, and varying R2 from 100 Ω to 5 Ωk to simulate varying thorax impedances (see Figure 4).

As can be seen in Figure 5, the current source works for impedance values from 1 Ω to approximately 2 Ωk. Outside of that range, the source is unable to provide a stable current. Even so, these values were sufficient for system functioning, as the maximum thoracic impedance that could be measured is 1 Ωk [15].

#### 2.4.3. Calibration

Despite the fact that the chip provides a recommended circuit to acquire impedance data and perform a calibration, that configuration works with a voltage excitation signal and not with a current source. This implies that an extra calibration step is needed, using the proposed circuit [4].

A Hioki IM3536 LCR (inductance (L), capacitance (C), and resistance (R)) Meter (Hioki inc., Ueda, Nagano Prefecture, Japan) was used for calibration to improve the accuracy of the measurement system.

Calibration is performed using the impedance set-up shown in Figure 4 varying R2 from 100 Ω to 5 kΩ on both channels.

Circuit test impedance was measured with the LCR impedance analyzer equipment in the same configuration as the TEB device: a current of 706.8 uA${}_{RMS}$ at 50 kHz. The data points were used to determine the interpolation line for both channels. Figure 6 shows the equation converting digital values to real impedance values, which was the same for both channels. An unexpected measure was obtained during the calibration process, this could be attributed to the AD5933. However, the obtained regression equation fits almost perfectly to the other measurements used.

## 3. Acquisition and Processing

### 3.1. Volunteer Enrollment

The inclusion criteria for the volunteers were healthy, between 18 and 30 years old, and with a body mass index under 30 (underweight, normal, or overweight). The exclusion criteria were no history of cardiac problems and no pacemakers or any chest-implanted device, as the device injects an alternating current through the electrodes placed in the chest. The study protocol was approved by the Ethics Committee of Universidad de Concepción CEEBE-435-2019) and was conducted in accordance to the Declaration of Helsinki. Written informed consent was obtained from all participants in the study.

Fifteen healthy volunteers—7 women and 8 men—were included in the study, with an age range of [21 to 25] years, mean (±SD) weight of 62.7 ± 12 kg, and mean height of 166 ± 10 cm.

### 3.2. Measurement Protocol

After the informed consent process, to perform the measurements on the subjects, the set-up shown in Figure 7 was prepared as follows.

Uncover the thorax of the volunteer.Clean the area where the four electrodes will be positioned with alcohol-soaked cotton to eliminate any traces of creams and grease.Two electrodes were symmetrically placed in the left ventral chest and two in the right ventral chest (Figure 7).Sit the volunteer in a chair.Place the Velcro belt with an elastic resistive band SS5LB (Biopac Systems Inc, Goleta, CA, USA) to record the respiratory movement, and to validate TEB estimation of respiratory frequency (see Figure 8) using a Biopac MP36 data acquisition unit (Biopac Systems Inc, Goleta, CA, USA).Start the measurement. In the first 30 s, breathe normally, then perform maximum inspiration, and then a maximum expiration (like a spirometry measurement).Continue breathing normally for another 20 s.Exchange measurement channels without moving the electrodes and reverse the channel order to ensure channel-independent TEB measurements.Repeat measurement, as indicated in step 7.Disconnect the TEB system and all electrodes from the subject.

Finally, the data are saved in a PC with Bluetooth connectivity for processing and validation of the respiratory frequency estimation.

### 3.3. Signal Processing

The impedance signals present noises attributed to the electronic components of the circuit and the impedance calculation error induced by the AD5933. A sample signal is shown in Figure 9.

According to the authors of [46], respiration is in the range of 0.1 to 0.5 Hz, equivalent to 6 and 30 breaths per minute (BPM), respectively. Therefore, to attenuate the high-frequency noise present in the measurements, a low-pass FIR filter with a cut-off frequency of 0.5 Hz and a gain of −65 dB at 2 Hz is used.

Then, to facilitate breathing rate detection, all signals are normalized after acquisition. All signals entering the breathing rate algorithm have zero mean and a standard deviation of 1 Ω (see Figure 10).

## 4. Results

### 4.1. Bioimpedance Measurements

Regarding the TEB values obtained from the measurements in the test subjects, Table 1 shows the values of the mean ($\bar{x}$), standard deviations (σ), and the minimal (Min) and maximal (Max) values obtained in Ω, separated by right and left hemithorax.

Figure 11 shows the raw TEB value distribution from all subjects for right and left hemithorax.

The typical distribution of the raw TEBs for each subject is shown in Figure 12. Only subjects 2, 3, and 8 have higher impedance on the right hemithorax. All other subjects have a higher impedance on the left hemithorax.

### 4.2. Respiratory Rate Validation

To validate that the acquired TEB signal can be used, the respiratory cycle calculation is compared to the data acquired from the elastic resistive band on the chest using the peak detection method “findpeaks” available in MATLAB R2017b (The MathWorks, Inc., Natick, MA, USA) (see Figure 13).

The respiratory frequency of each hemithorax is compared to the resistive band reference in breaths per minute (BPM) to generate a Bland–Altman plot, shown in Figure 14.

The Bland–Altman plots of the right and left hemithorax show all the samples (or respiratory cycles detected) within the limits of agreement. The system is able to measure the respiratory frequency with an average error of −0.18 ± 1.42 BPM in the right hemithorax and −0.52 ± 1.31 BPM for the left hemithorax.

## 5. Discussion

A device capable of measuring bioimpedance variations during the respiratory cycle through two channels is designed and implemented based on the AD5933 impedance analyzer. This bioimpedance increases with inhalation and decreases with the exhalation, which is as expected as the thorax expansion implies a higher impedance to the current flow. The variation range registered are approximately 1 Ω in normal breaths and 6 Ω in maximum inhalation and exhalation. These variations, despite being small, are well identified by the system.

Regarding the use of the AD5933, the calculation error induced by its internal Fourier transform algorithm must be taken into account. Although the phase of the impedance can be obtained, for this work, it was not considered, as the main objective of the device is to measure the variation in the TEB signal magnitude.

It should be noted that the measurement device works for impedance measurements ranging from 1 Ω to approximately 1800 Ω. Although the range is low compared to other impedance measurement equipment, it is more than enough to measure TEB as the maximum values do not exceed 650 Ω. Furthermore, a VCCS was developed and tested before the measurements, with good performance between the expected impedances.

The acquisition system was designed to work with rechargeable LiPo batteries of 3.7 V/800 mAh and with a average power consumption of 55 mAh the device can measure continuously during almost 15 h. Using wireless communication makes it a portable system, facilitating its use as a Point of Care (POC) device that can be brought to home visits or be available in out-of-hospital care facilities.

The electrodes’ location are essential to carry out the measurements since the variation of bioimpedance varies according to the structures. If they are misplaced, the change in impedance may not be acquired due to inhalation and exhalation, but instead from some other structure or organ, such as the diaphragm. In this case, the electrodes were located upper the zone of opposition of the thorax to avoid measuring diaphragm. A bipolar configuration was preferred as it requires fewer electrodes, which reduces the user’s discomfort. There were no differences regarding the shape of the impedance signal with the used configuration, but there were differences regarding its amplitude.

The highest impedance was obtained on the left side of the thorax (446.4 ± 2.3 Ω), compared to the right side of the thorax (437.4 ± 2.6 Ω). This behavior is not hardware-induced as the measurement protocol involved switching the channels and acquiring a second record for each subject. The influence of the heart in the left hemithorax or the fact that the right lung has three lobes and is slightly larger than the left lung with two lobes could explain these results. Notably, only three of the 15 subjects had a higher average impedance on the right side of the body (see Figure 12). This may be due to various factors, such as the electrodes location, inadequate cleaning of the measurement area, or poor electrode adherence on the subject’s skin.

The device was validated by comparing the respiration cycles obtained from the TEB signals concerning the reference signal of movement of the rib cage, obtained using an elastic resistive band. These cycles were identified utilizing the peaks of each signal. However, another methodology could be used to obtain the cycles, such as obtaining signal crossings by an established threshold, which would give fewer errors in detection of respiratory cycles, and therefore better results, as the location of peaks could have a margin of error when detecting false positives. The differences in the pronounced transients between the resistive elastic band signals and the filtered impedance signals (see Figure 13) are the results of measuring different parameters—the first measures the thorax expansion changes. The second is the change in the composition of the thorax’s internal structures—so the signal’s waveforms do not necessarily have to be the same.

The acquired signal allows estimating lung volume parameters related to the mechanical expansion and retraction of the thorax, which could allow estimating other physiological parameters such as tidal volume [47]. The proposed TEB device identifies the lung variations from both the right and left ventral sides of the thorax and can measure the respiratory frequency (Figure 14).

Although subjects of both genders, different ages, weights, and heights were measured, their characteristics are quite similar, so it cannot be yet guaranteed that the system works for subjects with other physical characteristics. For example, in the case of a volunteer with abundant body hair (Subject 4 in Table 1), the presence of excess hair generated more noise compared to the rest of the subjects, affecting the morphology of the acquired signal, and the identification of respiratory cycles and their correlation between hemithorax. The above is solved by shaving the chest area.

This device was not validated for measurements longer than one minute. However, during the duration of the tests, the measurements remained stable. On the other hand, none of the volunteers felt uncomfortable while taking the measurements. The only problem was regarding the placement and subsequent removal of the electrodes on the chest for people with much hair, which can be painful. Again, this issue can be solved by shaving the area where the electrodes will be placed.

## 6. Conclusions

This low-cost, portable TEB device could assist healthcare providers by detecting low correlations between the TEB signals of the right and left sides of the body or deviation from normal values. This development could be used as a point of care (POC) device to screen lung abnormalities such as pneumothorax, emphysema, or pneumonia. It could also provide a continuous monitoring method for lung state, avoiding trips and costs associated to Computerized Tomography (CT) or Magnetic Resonance Imaging (MRI) exams. To achieve this, a clinical validation with a larger sample is necessary, but preliminary results are promising. 

## Figures and Tables

**Figure 1 sensors-20-05228-f001:**
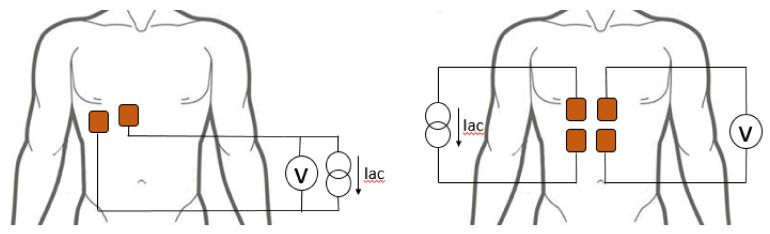
Configuration of electrodes for the alternate current injection (Iac) and the acquisition of impedance pneumography (IP) in a bipolar (**left**) and tetrapolar (**right**) configuration.

**Figure 2 sensors-20-05228-f002:**
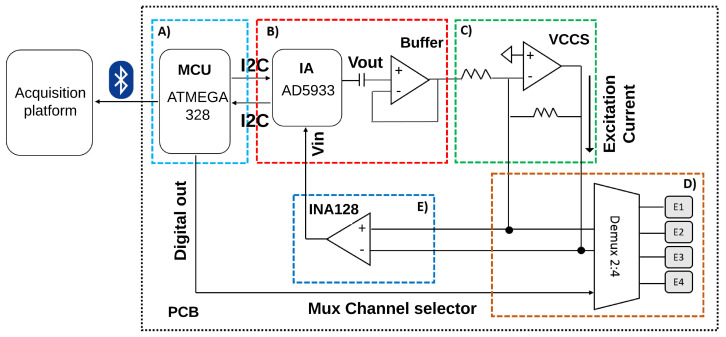
General diagram of the thoracic electrical bioimpedance (TEB) measurement system. (**A**) Control stage. (**B**) Generation of excitation voltage. (**C**) Conversion to excitation current. (**D**) Distribution. (**E**) Acquisition of differential voltage.

**Figure 3 sensors-20-05228-f003:**
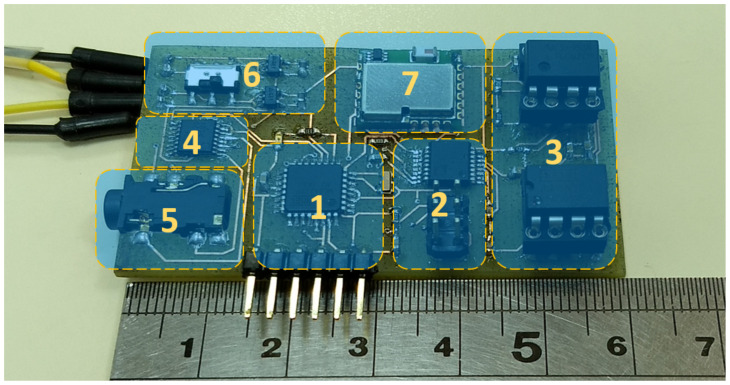
TEB device board, 60 × 30 mm [36].

**Figure 4 sensors-20-05228-f004:**
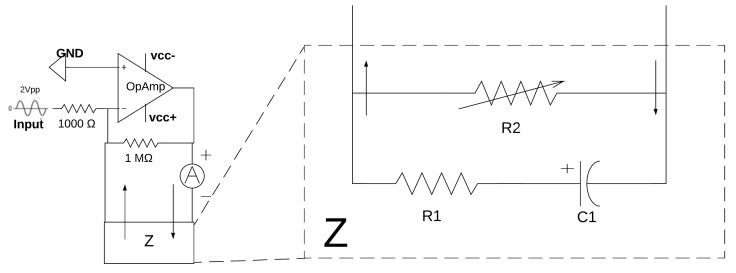
Impedance set-up for the voltage-controlled current source (VCCS) testing, with Z = Impedance.

**Figure 5 sensors-20-05228-f005:**
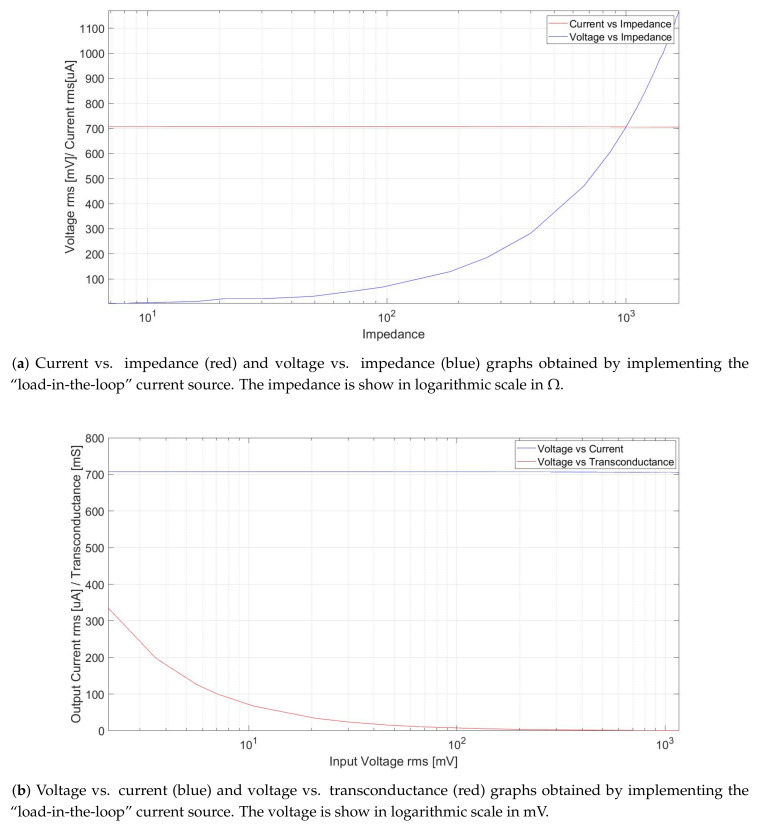
Results from the VCCS testing in a single frequency measurement system at 50 kHz. (**a**) The input voltage compensations due the impedances changes and (**b**) the transconductance changes of the implemented VCCS.

**Figure 6 sensors-20-05228-f006:**
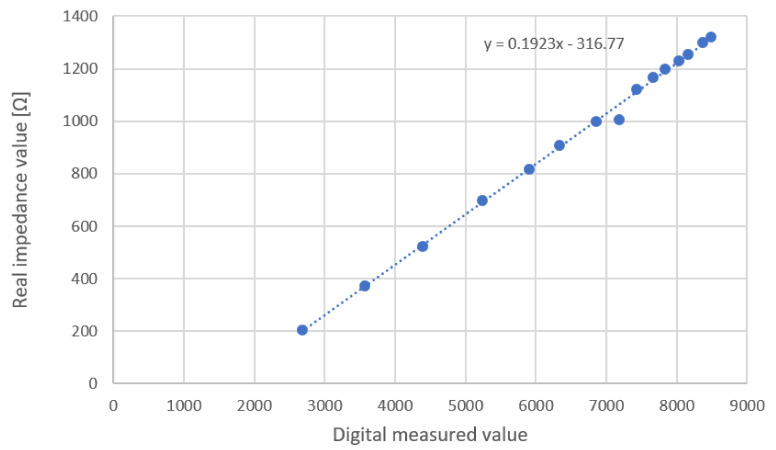
Calibration of TEB device channels. In the equation, *x* is digital measured values and *y* is impedance in [36].

**Figure 7 sensors-20-05228-f007:**
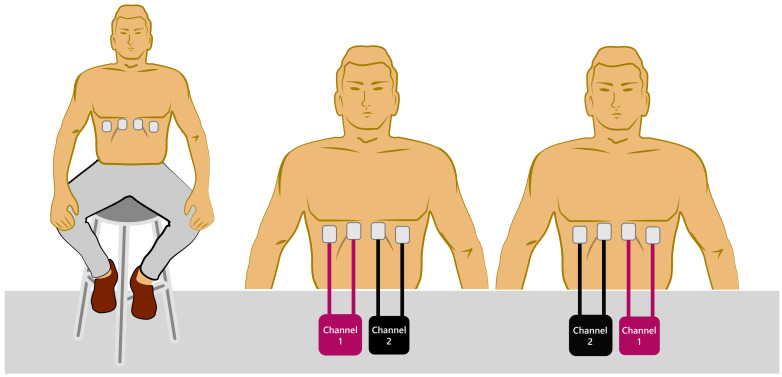
Volunteer position and channel configurations for TEB system measurements [36].

**Figure 8 sensors-20-05228-f008:**
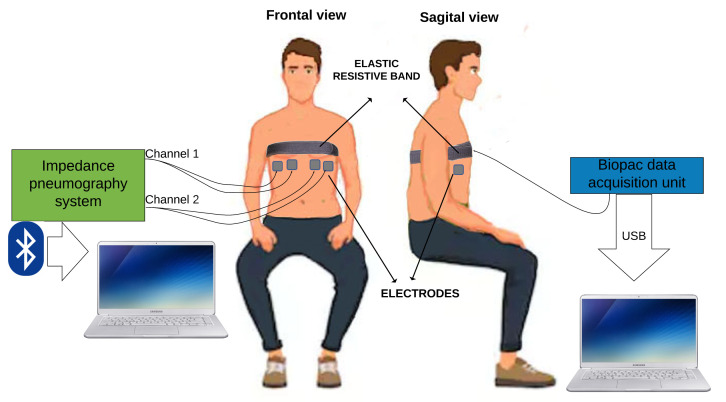
Electrodes and elastic resistive band locations during measurements.

**Figure 9 sensors-20-05228-f009:**
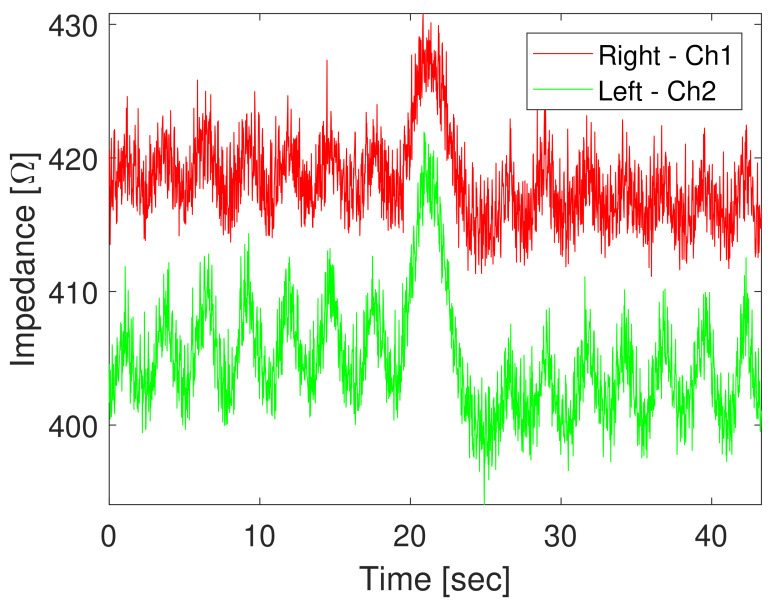
Example of raw TEB acquisition in a single subject [36].

**Figure 10 sensors-20-05228-f010:**
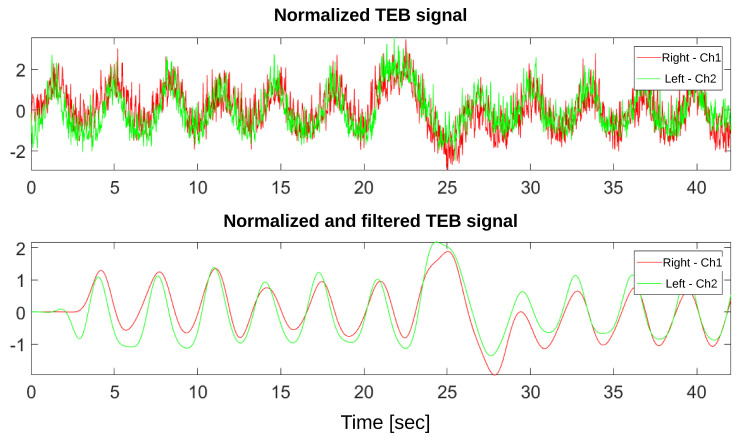
Normalized and unfiltered TEB variation signals from both channels (upper) and filtered and normalized TEB variation signals from both channels (lower).

**Figure 11 sensors-20-05228-f011:**
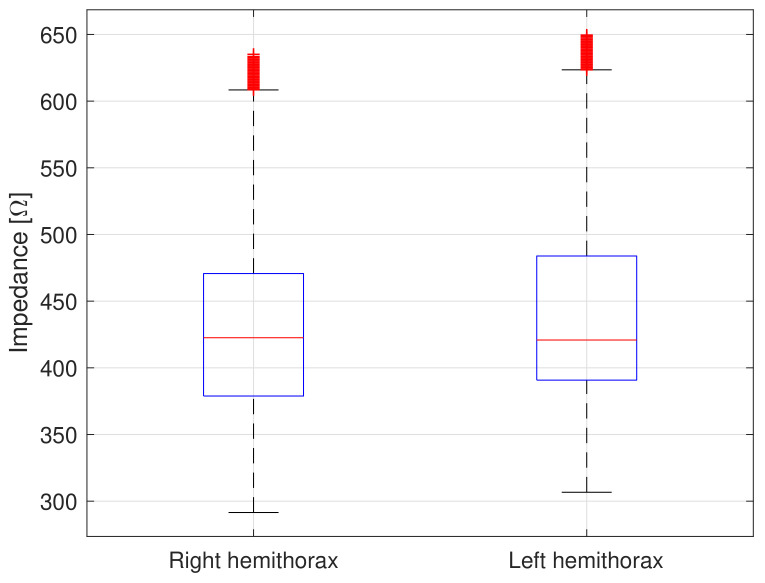
Boxplot of TEB values distribution for right and left hemithorax from all subjects [36].

**Figure 12 sensors-20-05228-f012:**
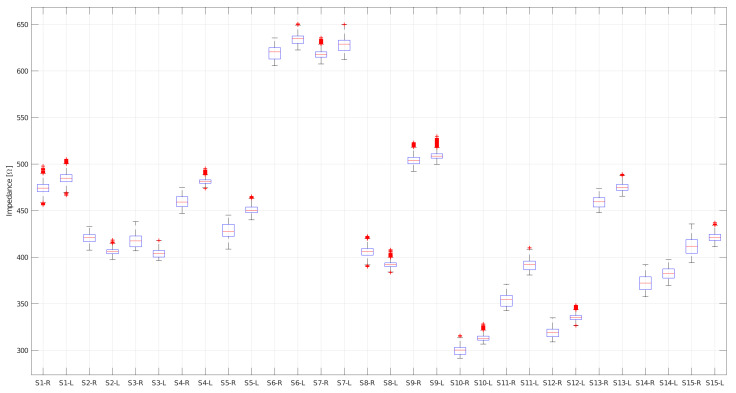
Distribution of raw TEBs for left (L) and right (R) hemithorax of all 15 subjects [36].

**Figure 13 sensors-20-05228-f013:**
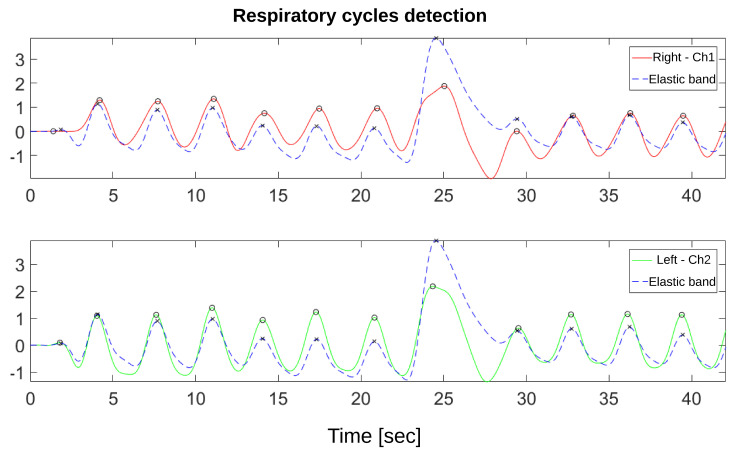
Identification of respiratory cycles using peak detection. Top image: channel 1 TEB variation signal (red) with detected peaks marked with “O”, and resistive band signal (blue) with peaks marked with “X”. Bottom image: channel 2 TEB variation signal (green) with detected peaks marked with “O”, and resistive band signal (blue) with peaks marked with “X”.

**Figure 14 sensors-20-05228-f014:**
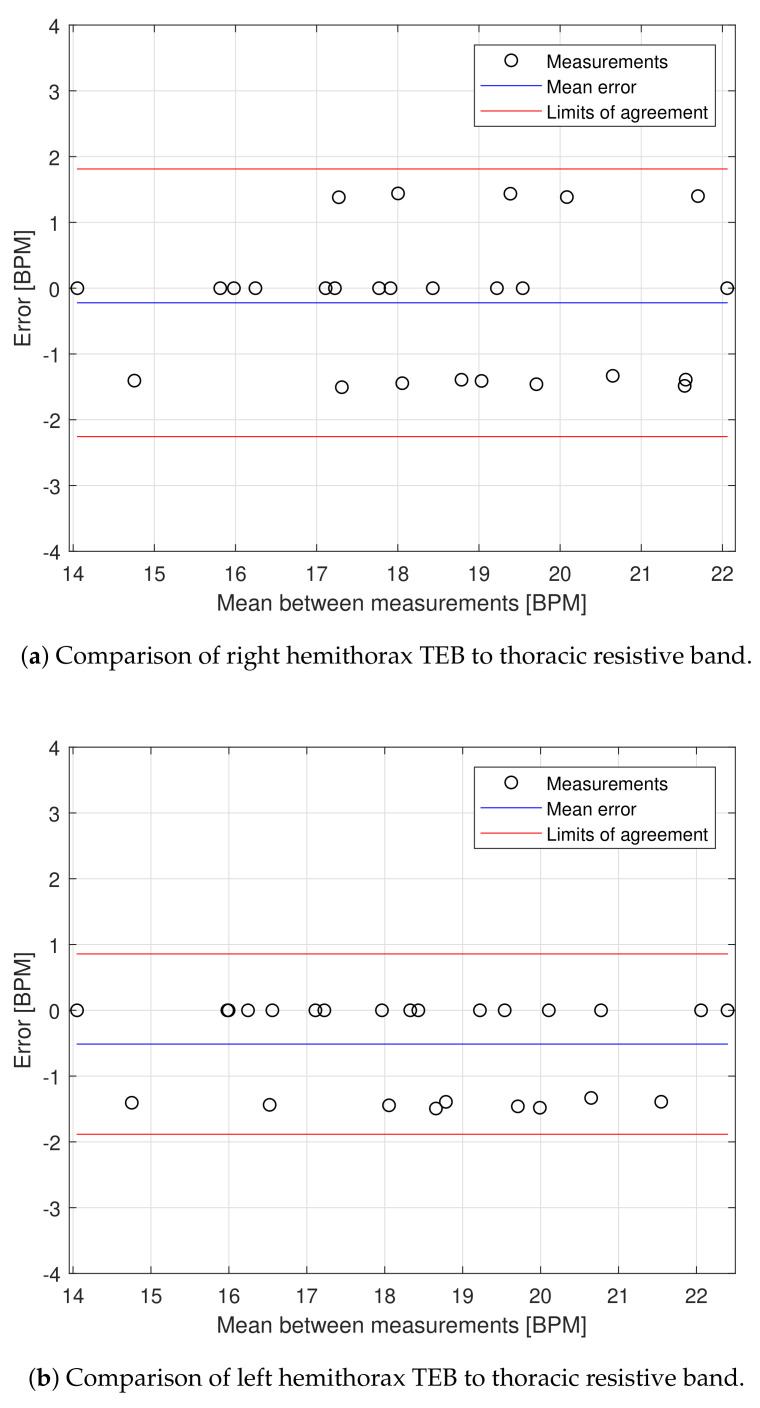
Bland–Altman plots for breaths per minute (BPM) estimation [36].

**Table 1 sensors-20-05228-t001:** TEB values obtained from signals analysis by channel and hemithorax. Subject X${}_{Ch1}$ means that the Channel 1 measures the right hemithorax and Channel 2 measures the left hemithorax, while Subject X${}_{Ch2}$ means that the Channel 2 measures the right hemithorax and Channel 1 measures the left hemithorax.

	Right Hemithorax				Left Hemithorax			
Subject	$\bar{x}$ [Ω]	[Ω]	Min [Ω]	Max [Ω]	$\bar{x}$ [Ω]	[Ω]	Min [Ω]	Max [Ω]
Subject 1${}_{Ch1}$	475.5	2.6	470.3	484.4	487.4	1.8	483.7	493.2
Subject 1${}_{Ch2}$	473.2	2.7	668.0	482.0	482.2	1.4	479.4	486.6
Subject 2${}_{Ch1}$	424.3	1.8	419.4	428.9	405.7	2.2	402.1	411.6
Subject 2${}_{Ch2}$	416.7	2.4	409.8	424.7	406.0	3.1	399.0	413.4
Subject 3${}_{Ch1}$	423.2	2.8	419.8	434.1	407.4	1.5	405.4	414.8
Subject 3${}_{Ch2}$	411.1	0.8	409.6	414.5	400.2	0.6	398.8	401.8
Subject 4${}_{Ch1}$	464.9	2.0	461.1	471.1	481.8	2.3	478.8	490.9
Subject 4${}_{Ch2}$	454.3	1.3	451.5	458.1	480.8	2.3	477.6	489.9
Subject 5${}_{Ch1}$	433.2	5.1	419.7	440.2	453.5	2.5	448.7	460.1
Subject 5${}_{Ch2}$	422.8	4.4	411.1	429.2	447.8	1.8	444.3	453.8
Subject 6${}_{Ch1}$	613.3	3.0	607.9	622.7	630.5	3.7	626.2	644.4
Subject 6${}_{Ch2}$	625.0	2.3	620.1	632.4	637.4	1.9	635.1	647.4
Subject 7${}_{Ch1}$	618.8	2.8	614.2	627.2	622.0	2.9	616.9	631.1
Subject 7${}_{Ch2}$	616.7	3.7	611.1	631.4	633.3	2.5	629.9	644.7
Subject 8${}_{Ch1}$	409.5	3.1	405.6	419.3	393.5	2.3	389.4	401.8
Subject 8${}_{Ch2}$	402.7	2.6	399.5	412.3	390.5	2.7	386.9	400.1
Subject 9${}_{Ch1}$	507.1	3.3	501.7	519.4	510.8	3.5	506.3	525.3
Subject 9${}_{Ch2}$	500.9	3.2	495.6	512.1	507.6	3.7	504.2	521.3
Subject 10${}_{Ch1}$	303.2	2.0	300.9	311.6	315.2	2.2	312.0	323.7
Subject 10${}_{Ch2}$	296.0	1.9	294.1	305.2	311.7	2.2	309.8	321.5
Subject 11${}_{Ch1}$	359.0	1.9	356.4	366.2	395.8	2.3	392.3	405.2
Subject 11${}_{Ch2}$	347.8	1.8	346.1	356.3	386.9	2.0	385.1	396.7
Subject 12${}_{Ch1}$	322.6	2.1	319.2	381.6	336.9	1.9	334.1	345.3
Subject 12${}_{Ch2}$	315.2	1.9	312.6	324.3	333.6	1.8	331.4	342.3
Subject 13${}_{Ch1}$	463.6	2.1	459.8	470.3	477.7	2.5	472.6	486.2
Subject 13${}_{Ch2}$	454.2	1.5	452.0	461.5	472.2	2.5	468.4	483.1
Subject 14${}_{Ch1}$	378.9	3.6	373.1	386.8	387.3	2.7	382.5	392.6
Subject 14${}_{Ch2}$	365.2	2.6	361.2	372.6	377.3	1.8	374.2	383.3
Subject 15${}_{Ch1}$	418.8	4.4	411.4	431.7	424.2	3.1	419.3	432.8
Subject 15${}_{Ch2}$	404.8	3.6	399.1	418.6	418.2	2.4	414.8	427.3
$\bar{x}$	437.4	2.6	438.3	446.4	445.7	2.3	442.2	454.45

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
