# Peer review of "Simple Wireless Impedance Pneumography System for Unobtrusive Sensing of Respiration"

_sensors, 2020, doi:10.3390/s20185228_

Round 1

Reviewer 1 Report

Presented simple wireless system for unobtrusive sensing of respiration confirms importance of the promising method of impedance pneumography, which while known for a long time is still undervalued.  Comments:

  • While it is stated that “many researchers” have been in involved in development of bioimpedance devices only very few are referenced. It could be expanded and detailed a bit more.
  • Main safety standard for medical apparatus is not mentioned. Some statements require references, such as “frequency of 50 kHz” is “commonly used” and that it “does not stimulate electrically excitable tissue such as nerves or heart muscles”. What about analysis of the failure conditions of the equipment?
  • What about sensitivity distribution? What kind of impact could it have? What about linearity of the relationship between impedance and breathing volume? Why exactly those chosen locations and distances?
  • Excitation signal generation. What does guarantee that there is “no offset”? What about failure conditions? What happens if there is offset?
  • VCCS testing. Why is there straight line on the voltage vs impedance curve (fig. 5) between ca 300 and 400 ohms?
  • Why there is (only) one strong outlier on fig. 6?
  • Signal processing. Is 0,5 Hz optimal? What drawbacks could it have?
  • Page 10, line 171-173: why?
  • 13: why the pronounced strong difference between band and impedance during transient?
  • The electrode locations are mentioned again, but not substantially justified. Between line 206-211: what about long-term stability? Also related to the previous question lines 234-237 require substantially better justification. How long can the monitoring continue with current electrodes? Why was the system limited to 6-30 BPM?

Generally nice and interesting work.

Author Response

Dear Reviewer,

We appreciate the comments and suggestions sent. We detail below the modifications that have been made respecting the order of the comments submitted:

  1. The introduction section was rewritten, incorporating the recommendations, justifying the statements made with previously published works and books.
  1. In the second paragraph of section 2.1, safety considerations are rewritten, explaining not only current intensity but also frequency. This was justified with the article number 37 in our reference section.
  2. The last paragraph of section 2.2 justifies the use of a bipolar configuration to measure impedance. No comments can be made regarding the linearity relationship between measured impedance and measured air volume since tidal volume was not measured in this work.
  3. About the safety considerations the current source was tested between the impedance expected with good performance. We ensure that there is no offset in the impedance range tested. In the Discussion section in line 227 the performance of the current source implemented is mentioned.
  4. Figure 5 is modified in which a measurement was repeated due to a typing error, which explains the straight-line present between 300 and 400 ohms.
  5. Regarding the "outlier" present in Figure 6, between lines 151 and 153 we propose an explanation mentioning that "An unexpected measure was obtained during the calibration process, this could be attributed to the AD5933. However, the obtained regression equation fits almost perfectly to the other measurements used".
  6. Regarding the disadvantages of the filter used for processing impedance signals, we believe that the cutoff frequency of 0.5 Hz is appropriate, preserving oscillations due to respiration and filtering out high frequency components present in the acquired signals.
  7. In the Discussion section, line 245 incorporates suggestions about why we believe that some impedance measurements in some subjects did not follow the expected mean distribution.
  8. In the Discussion section in line 268 we mention that this device was not validated for measurements longer than one minute. However, during the duration of the tests, the measurements remained stable. We deleted the comment about the limitation of the system to 6-30 BPM, because this only referred to the filter used in the signal processing.
  9. The sentence in the Discussion section was deleted as we cannot properly justify it with references: “A direct use of this device would be for home monitoring of patients with cardiac failure, to detect dangerous fluid build-up. After having acquired a baseline, a developing pulmonary edema due to cardiac failure would be directly translated into a change in thoracic impedance, which could be detected with a simple daily measurement."
  10. The grammar and vocabulary of the paper were professionally reviewed.

Again, we appreciate the suggestions and hope that our modifications improved the quality and understanding of our manuscript, in order to comply with the standard of the journal.

Sincerely,

D.Sc. Esteban Pino Q.

Universidad de Concepción, Chile.

Edmundo Larenas 219, Concepción, Biobio, Chile

estebanpino@udec.cl

Tel: +56 9 81989266

Reviewer 2 Report

This article resents a device for bioimpedance measurements and its use for respiration monitoring. The article is interesting, although the application and instrumentation design do not demonstrate significant novelties.

  1. The introduction needs to be strengthened, especially the 1st paragraph. This statement is not valid and very general: "Also, bioimpedance used in medical applications is expensive, and other simpler alternatives are only available at a prototype level". Your device is also an academic prototype. i don't understand what you mean here. This statement: "The low cost, improvements in safety, and the development of better electronic components has encouraged the use of bioimpedance techniques in biomedical applications" must also be changed. It doesn;t say much, it is very general and the syntax/English are not very good. Please revise.
  2. Line 32: It would be worth mentioning here what this ΔR typically is or a range of values. Also what is the standard deviation from data reported in the literature for this 500 Ω base line resistance. Please add these. Are you referring here on DC values or AC values. If its AC please indicate the frequency.
  3. Impedance tomography is also often used for monitoring respiration and lung content with draeger pulmovista being one commercial example: https://www.draeger.com/en_aunz/Hospital/EIT-Lung-Monitoring. For completeness you should also discuss EIT and mention what is the advantage simple single 2 or 4 point measurements have over EIT systems.
  4. In terms of instrumentation, many groups have used the AD5933 chip, which is rather dated. There are other commercial components like the AFE4300 and other chips from texas instruments specifically designed for respiration such as the AFE1292R and AFE1298R etc. You should mention these other commercial alternatives and what have others used, developing custom bioimpedance instrumentation with commercially-available chips. Fully custom CMOS solutions should be also briefly mentioned for completeness (e.g. work from Demosthenous, Hyung-Joun Yoo or Odame).
  5. You should also discuss in the introduction further what is the clinical importance for respiration monitoring. In which clinical applications it is important to monitor breathing?
  6. Section 2.1 should be re-written. I suggest you read this paper: Lionheart, Kaipio and Mcleod, Generalized optimal current patterns and electrical safety in EIT, Phys. Meas, vol. 22, no. 1, 2001. Please have a read on the discussions on safety limits and currents as a function of frequency according to the relevant safety standards.
  7. Section 2.2: Regarding electrode configurations you are citing a technical note. It would be better if you cited some academic literature as well and also provide some additional discussions regarding what are the advantages of tetrapolar measurements. I suggest you to have a look at the Grimnes and Martinsen book, and publications from Kassanos.
  8. Please change the title for section 2.3 to Hardware design or instrumentation design or something else.
  9. Please provide more details about the developed circuit. Please show which terminals of the AD5933 have been used to collect each component. Please provide information about all the components used. What does AO stand for in Fig. 2? Which component did you use for this OpAmp? Which MUX did you use? Are leakage and parasitics in the MUX introducing any errors? Please provide values for all passive components used, as well as the bluetooth module used and voltage regulators.
  10. Why have you used a load-in-the loop approach? Please discuss. Why not a more standard balanced current source like a modified Howland? Ideally you should provide characterization results regarding the output impedance and output current of the proposed current source as a function of frequency. Nevertheless, I understand that you are only interested in the performance at a single frequency, 50 kHz. What is the transcondutance of the current source? Please report it. You mention the input voltage and the output current, but please also mention the transconductance as well.
  11. Testing of the current source should also include electrode contact impedance models for the two electrodes not only tissue impedance models. The greatest challenge for the current source is the large electrode impedance. This model should correspond to the impedance of the Ag/AgCl electrodes you used. It is possible that at 50 kHz their electrode is very small, because they are probably large and also because they are non-polarizable, but this should be demonstrated and discussed, if it is indeed the case. Otherwise, VCCS testing should include these 2 impedances.
  12. Please highlight that the results of Fig. 5 are at 50 kHz and also highlight why, i.e. that this is a single frequency measurement system.
  13. You need to provide further discussion as to why 50 kHz have been used only. This is typically guided mainly by the physiological application, i.e. which frequency provides higher sensitivity and more information.
  14. I do not understand how and why the results of Fig. 6 where obtained. Please provide more information and discussions.
  15. This statement is not very accurate: "Although the phase of the impedance can be obtained, for this work, it was not considered, since the main goal for the device is to measure the variation in the TEB
    signal." Bioimpedance includes both magnitude and phase. I think it is better if you change this into: "Although the phase of the impedance can be obtained, for this work, it was not considered, since the main goal for the device is to measure the variation in the TEB signal magnitude."
  16. What is the power consumptio of the system? How long can the system operate with the batteries used. What are the battery specs?
  17. What electrodes did you use? Please provide manufacturer and model details.
  18. Please improve the English here: "A solution to this would be to shave the area, which may rejected by the subject. " Line 228-229.
  19. Conclusions should be rewritten. Currently it discusses potential future developments with no evidence.
  20. Please significantly highlight the novelty of the paper.

Author Response

Dear Reviewer,

We appreciate the comments and suggestions sent. We detail below the modifications that have been made respecting the order of the comments submitted,

  1. The introduction section was rewritten, incorporating the recommendations, justifying the statements made with previously published works and books. In particular:
    • In line 27, we mention and explain what electrical impedance tomography is, indicating its advantages and disadvantages concerning impedance pneumography.
    • In line 35, the characteristic values of thoracic impedance signals are justified with previous studies.
    • In line 48 some clinical applications of electrical impedance pneumography measurement systems are discussed.
    • In line 52, other integrated devices are mentioned in addition to AD5933 that have been used to develop systems for monitoring respiration with electrical impedance, indicating their advantages and disadvantages.

2. In the last paragraph of the introduction, this paper's objective is indicated, reporting the main steps carried out in this development, the challenges and considerations taken into account, and the main characteristics of thoracic impedance obtained in our measurements.In the second paragraph of section 2.1, safety considerations are rewritten, explaining not only current intensity but also frequency. This was justified with the inclusion of article [37] in our references.

3. The last paragraph of section 2.2 in line 94-96 justifies the use of a bipolar configuration to measure impedance.

  1. The title of section 2.3 is modified to Hardware design as suggested, in line 97.
  2. The information in Figure 2 was complemented by indicating the communication pins and the excitation signal's conditioning elements. In section 2.3 all components used are specified, also indicating the company that manufactures them.
  3. Section 2.3.2 mentions that there are other types of voltage-controlled current sources, and the use of the source used in this development is justified. Our intention is not to obtain precise impedance measurements for the generation of images but to obtain the variations of the same and extract characteristics that could, in the future, provide important parameters using a simplified system.
  4. Unfortunately, we couldn´t test electrode contact impedance models in the times of the reviewed version due to the sanitary context, as our University has entered a self-quarantine mode and we are not allowed into the laboratory.
  5. Figure 5 had a typing error repeating a measurement, which explains the straight-line present between 300 and 400 ohms. We corrected that error in this new version. Also, the information obtained is complemented by a graph that presents the transconductance of the current source. The caption of the figure indicates that these results are measurements at a single frequency.
  6. In the Introduction section line 36 we justify the frequency used for thoracic impedance measurements.
  7. In line 148-153 we provided more information about the calibration process.
  1. We changed the statement as required, in line 221.
  2. The discussion section In line 229 explains the autonomy of the system using batteries.
  3. In section 2.3.2 in line 111, we provided electrode manufacturer information.
  4. In line 267 the phrase "A solution to this would be to shave the area, which may be rejected by the subject." is replaced for "The above is solved by shaving the chest area."
  5. The discussion and conclusion of the work were rewritten to give a better structure to the paper, in lines 213-272.
  1. We highlight the novelty of the paper in the last paragraph in introduction section, in lines 61-69.
  2. The grammar and vocabulary of the paper were professionally reviewed.

Again, we appreciate the suggestions and hope that our modifications improved the quality and understanding of our manuscript, in order to comply with the standard of the journal.

Sincerely,

D.Sc. Esteban Pino Q.

Universidad de Concepción, Chile.

Edmundo Larenas 219, Concepción, Biobio, Chile

estebanpino@udec.cl

Tel: +56 9 81989266

Reviewer 3 Report

This letter is extended upon the authors’ EMBC conference presentation. A prototype-level portable wireless system for thoracic electrical bioimpedance monitoring is designed, and some meaningful test results are provided. I think that the presented work is solid and the manuscript is generally well written. I think this letter can be accepted pending some minor revisions. - I think that the Introduction should be enhanced. The current issue, other approaches to solve the issues, your idea, surrounding current researches around your idea, your originality, and research objectives are not clear at all. - This work does not consider image reconstruction of the lung region using the electrode measurement. I think that by sensing and learning the spatiotemporal structural signatures in the impedance distribution [Ref1], more information can be extracted. The authors are suggested to at least discuss this point in an appropriate place. [Ref1] Time sequence learning for electrical impedance tomography using Bayesian spatiotemporal priors, IEEE Transactions on Instrumentation and Measurement, vol. 69, no. 9, pp. 6045–6057, Sept. 2020.

Author Response

Dear Reviewer,

We appreciate the comments and suggestions sent. We detail below the modifications that have been made respecting the order of the comments submitted,

  1. The introduction section was rewritten, incorporating the reviewer's recommendations, justifying the statements made with previously published works and books. In particular:
    • In line 27, we mention and explain what electrical impedance tomography is, indicating its advantages and disadvantages concerning impedance pneumography.
    • In line 35, the characteristic values of thoracic impedance signals are justified with previous studies.
    • In line 48 some clinical applications of electrical impedance pneumography measurement systems are discussed.
    • In line 52, other integrated devices are mentioned in addition to AD5933 that have been used to develop systems for monitoring respiration with electrical impedance, indicating their advantages and disadvantages.
    • In the last paragraph of the introduction, this paper's objective is indicated, reporting the main steps carried out in this development, the challenges and considerations taken into account, and the main characteristics of thoracic impedance obtained in our measurements.
  1. In the introduction, line 39, when talking about impedance by pneumography, this statement was incorporated with the related work: This work does not consider image reconstruction of the lung region using the electrode measurement.
  2. The discussion and conclusion of the work were rewritten to give a better structure to the paper, in lines 213-272.
  3. The grammar and vocabulary of the paper were professionally reviewed.

Again, we appreciate the suggestions and hope that our modifications improved the quality and understanding of our manuscript, in order to comply with the standard of the journal.

Sincerely,

D.Sc. Esteban Pino Q.

Universidad de Concepción, Chile.

Edmundo Larenas 219, Concepción, Biobio, Chile

estebanpino@udec.cl

Tel: +56 9 81989266

Round 2

Reviewer 2 Report

The English can be improved more.